# First Isolation and Identification of *Aeromonas veronii* in a Captive Giant Panda (*Ailuropoda melanoleuca*)

**DOI:** 10.3390/ani13172779

**Published:** 2023-08-31

**Authors:** Xiaoyan Su, Mei Yang, Yunli Li, Xia Yan, Rong Hou, James Edward Ayala, Lin Li, Chanjuan Yue, Dongsheng Zhang, Songrui Liu

**Affiliations:** Chengdu Research Base of Giant Panda Breeding, Sichuan Key Laboratory of Conservation Biology for Endangered Wildlife, Sichuan Academy of Giant Panda, Chengdu 610081, China; xyansu@126.com (X.S.); yangmei20230606@163.com (M.Y.); liyunli1007@163.com (Y.L.); 17311495120@163.com (X.Y.); hourong202306@163.com (R.H.); giantpanpan666@gmail.com (J.E.A.); lilin_yx@163.com (L.L.); chanjuan_yue@163.com (C.Y.); dongsheng930206@163.com (D.Z.)

**Keywords:** *Aeromonas veronii*, giant panda, antibiotic resistance, pathogenicity, pathology

## Abstract

**Simple Summary:**

*Aeromonas veronii*, an important aquatic bacterium, has been known to infect a variety of animals worldwide. However, there have been no reports of *A. veronii* infection in giant pandas. Here, we found that strain VGP was resistant to six antibiotics, carries a large number of resistance genes and virulence genes, and has strong pathogenicity in mice. These results suggest that infection with strain VGP could be one of the leading causes of death for this giant panda cub. Our study presents the first evidence that giant pandas could be infected by *A. veronii*.

**Abstract:**

The objective of this study was to understand biological characteristics of one bacteria strain named as VPG which was isolated from multiple organs of a dead captive giant panda cub. Here, we use biochemical tests, 16S rRNA and *gyrB* genes for bacterial identification, the disk diffusion method for antibiotic resistance phenotype, smart chip real-time PCR for the antibiotic resistance genotype, multiplex PCR for determination of virulence genes, and the acute toxicity test in mice for testing the pathogenicity of isolates. The isolate was identified as *A. veronii* strain based on the biochemical properties and genetic analysis. We found that the strain carried 31 antibiotic resistance genes, revealed antimicrobial resistance phenotypically to several antibiotics including penicillin, ampicillin, oxacillin, amoxicillin, imipenem, and vancomycin, and carried virulence genes including *aer*, *act*, *lip*, *exu*, *ser*, *luxs*, and *tapA*. The main pathological changes in giant panda were congestion, necrotic lesions and a large number of bacteria in multiple organs. In addition, the LD_50_ in Kunming mice infected with strain VGP was 5.14 × 10^7^ CFU/mL by intraperitoneal injection. Infection with strain VGP led to considerable histological lesions such as hemorrhage of internal organs, necrosis of lymphocytes and neurons in Kunming mice. Taken together, these results suggest that infection with strain VGP would be an important causes of death in this giant panda cub.

## 1. Introduction

*Aeromonas veronii*, a Gram-negative, rod-shaped, mesophilic, facultative anaerobic bacterium, is widely distributed in the aquatic environment [1,2]. As a common pathogen in aquaculture, *A. veronii* can infect a variety of aquatic animals, including sea bass (*Lateolabrax maculatus*), crucian carp (*Carassius auratus gibelio*) [3,4], Chinese alligator (*Alligator sinensis*) [5], Yangtze finless porpoise (*Neophocaena asiaeorientalis*) [6] and soft-shelled turtle (*Pelodiscus sinensis*) [7] and clinical symptoms are skin ulcers and visceral hemorrhages in these species. *A. veronii* can also cause visceral hemorrhage in ducks (*Anatinae*) [8]; diarrhea, lethargy, anorexia, and visceral hemorrhage in foxes (*Vulpes* sp.) [9]; diarrhea and hepatomegaly in wild yak (*Bos mutus*) [10]; coughing, sneezing and sniveling in rabbits (*Leporidae* sp.) [11]; and disease of Iberian lynx (*Lynx pardinus*) [12]. It may be a potential pathogen of tawny owl (*Strix aluco*) and scarlet ibis (*Eudocimus ruber*) [13,14]. More seriously, it could cause gastroenteritis, bacteremia/septicemia and respiratory tract infections in humans [15]. In addition, wild waterfowl may carry pathogenic *Aeromonas* species in their intestine; therefore, the migration of waterfowl is a potential mechanism for the global distribution of *Aeromonas* [16].

As an iconic flagship species for wildlife conservation, the giant panda (*Ailuropoda melanoleuca*) is considered a national treasure in China and is a Class 1 protected endemic species [17]. The giant panda is currently categorized as vulnerable by the International Union for Conservation of Nature (IUCN), and faces continued threats from habitat fragmentation and infectious diseases in both the in situ and ex situ populations. Meanwhile, bacterial diseases, such as *Escherichia coli* [18,19], *Klebsiella pneumoniae* [20], *Proteus mirabilis* [21], *Proteus vulgaris* [22], *Enterobacter cloacae* [23], and *Staphylococcus aureus* [24], are more and more reported in the giant panda. So far, there is no report of infection and biological characteristics of *A. veronii* in the giant panda.

The objective of the present study investigated the cause of death of captive giant panda cubs and analyzed the identification, antibiotic sensitivity, virulence gene, and pathogenicity of the *A. veronii* isolated from the internal organs of a dead captive giant panda cub.

## 2. Materials and Methods

### 2.1. Organ Sampling and Bacterial Isolation

One dead captive one-day-old giant panda cub with no clinical symptoms was sampled in 2020, in Sichuan province, China. Brittle texture of liver and spleen, dark red liver with hemorrhagic spots and flatulence, pulmonary congestion and hemorrhage, pericardial effusion were found in autopsy. The heart, liver, lung, and kidney were collected for bacterial examination as a routine check. The samples were cultivated on 5% (*v*/*v*) sheep blood agar at 37 °C for 24 h and the bacterial colonies were purified by streaking onto the sheep blood agar plate twice. A single bacterial colony was selected and inoculated in Brain Heart Infusion (BHI) medium at 37 °C for 24 h, and then preserved at −80 °C in the BHI medium containing 40% (*v*/*v*) sterile glycerol. The isolate was named as VGP (*Aeromonas veronii* in giant panda). 

### 2.2. Histopathology of Giant Panda

To clarify the histopathological changes in dead giant panda, histopathology was performed on the heart, liver, spleen, lung, and kidney under a light microscope (Leica DM4B optics) following 10% formalin fixation and hematoxylin–eosin staining.

### 2.3. Analysis of Physiological and Biochemical Features

Staining characteristics of isolate VGP heart, liver, lung, and kidney were studied by light microscope. Biochemical features of isolate were performed by microbiochemical reaction tube including oxidase, lysine decarboxylase, citrate, indole, mannitol, sucrose, maltose, V-P, cellobiose, H_2_S production, L-lactate alkalinization, L-lactate alkalinization, and glycine arylamidase. The biochemical reactions were performed in three parallel groups. The biochemical features had been identified according to the Berger’s Manual of Systematic Bacteriology.

### 2.4. Sequence Analysis of 16S rRNA and gyrB Genes

The genomic DNA of the isolate VGP were extracted using a TIANamp Bacterial DNA Kit (Tiangen-Biotech, Beijing, China) following the manufacturer’s instructions. Genomic DNA was stored at −20 °C for analysis. The classic PCR of 16S rRNA gene (27 F: 5′-AGAGTTTGATYMTGGCTCAG-3′, 1492 R: 5′-GGTTACCTTGTTACGACTT-3′, Product size: 1456 bp, Tm: 54 °C) and *gyrB* gene (3F: 5′-TCCGGCGGTCTGCACGGCGT-3′, 27R: 5′- TTGTCCGGGTTGTACTCGTC-3′, Product size: 1124 bp, Tm: 60 °C) were amplified by a pair of universal primers and *gyrB* gene primers [25,26]. Each PCR mixture was 25 µL in total volume consisting 12.5μL of Dream Taq Green PCR Master Mix (2×), 1 µL of each forward and reverse primer, 2 µL of template DNA and 8.5 µL of ddH_2_O. The reaction mixtures were subjected to start at 94 °C for 5 min prior to 30 cycles of amplification with 94 °C for 1 min, Tm for 1 min, 72 °C for 1 min; final extension at 72 °C for 10 min. The sequences analysis was performed by BLAST in NCBI (https://www.ncbi.nlm.nih.gov/ accessed on 22 January 2020.). The phylogenetic trees were established using the Neighbor-joining method in the MEGA 7.0 software package.

### 2.5. Molecular Identification of Virulence Genes

Virulence genes including *aer*, *act*, *ser*, *aha*, *lip*, *exu*, *luxs*, *tapA*, and *gcaT* were detected by classic PCR using primers in previously published studies [2,3] (Table 1).The PCR Products were sequenced by Sangon Biotech Co., Ltd. (Shanghai, China), and sequences were analyzed by BLAST in NCBI (https://www.ncbi.nlm.nih.gov/ accessed on 22 January 2020.). 

### 2.6. Antimicrobial Susceptibility Testing

Antimicrobial susceptibility testing was carried out by a disk diffusion method (K-B method) recommended by the CLSI 2020 [27]. Forty-two antibiotics (Hangzhou Microbiological Reagent Co., Ltd., Hangzhou, China) were chosen, including penicillin (PEN), piperacillin (PIP), ampicillin (AMP), oxacillin (OX), amoxicillin (AMX), moxalactam (MOX), ceftazidime (CMZ), cefepime (FEP), cefotaxime (CTX), cephalexin (CA), cefazolin (CZ), ceftriaxone (CTR), cefoxitin (FOX), piperacillin/tazobactam (TZD), cefuroxime (CXM), cefaclor (CEC), ampicillin/sulbactam (AMS), cefoperazone (CFP), ceftizoxime (ZOX), aztreonam (AT), meropenem (MEM), imipenem (IPM), kanamycin (K), gentamicin (GM), streptomycin (S), enoxacin (ENX), ofloxacin (OFX), norfloxacin (NOR), lomefloxacin (FOX), fleroxacin (FOX), levofloxacin (LVX), ciprofloxacin (CIP), gatifloxacin (GAT), chloramphenicol (C), vancomycin (VA), azithromycin (AZM), doxycycline (DX), minocycline (MI), trimethoprim-sulfamethoxazole (SXT), and trimethoprim (TMP). *E. coli* ATCC25922 was used as the quality control bacterial strain. The results were defined as susceptible (S), intermediate (I), and resistant (R) according to the CLSI 2020 breakpoints [27].

### 2.7. Molecular Identification of Antibiotic Resistance Genes (ARGs)

The genomic DNA of the isolate VGP were extracted using a TIANamp Bacterial DNA Kit (Tiangen-Biotech, Beijing, China) following the manufacturer’s instructions. Additionally, the total genomic DNA was used as the template. The Wafergen smart chip real-time PCR system was used to analyze the VGP chromosome antibiotic resistance genes with a total of 97 primer sets (see Appendix A) [28]. Each sample was repeated three times simultaneously. Following the initial activation of enzymes at 95 °C for 10 min, 30 cycles of the following procedure were used for amplification: denaturation at 95 °C for 30 s, and annealing at 60 °C for 30 s. Then, the results were analyzed using smart chip qPCR software to exclude the wells with multiple melting peaks or amplification efficiency beyond the range (90–110%). The calculation method is based on the previous study [28,29]. Briefly, the mapping data in the relative copy number = copy number (gene)/copy number (16S), in which the copy number (gene) and the copy number (16S) belong to the same sample. Copy number = 10 (33 − CT)/(10/3), i.e., if the offline data have a CT value of null, the CT value is 33, and the copy number value is 1.

### 2.8. The Pathogenicity Testing in Kunming Mice

Female Kunming mice (aged 6 to 8 weeks old) were purchased from the Chengdu Dossy Experimental Animals Co., Ltd. (Chengdu, China) and placed in polypropylene cages with sawdust bedding and kept under hygienic control. Forty-eight mice were randomly divided into 6 groups (8 mice per group). Bacterial isolate VGP were incubated overnight at 37 °C, and following the suspension in 0.9% endotoxin-free saline solution, animals were inoculated intraperitoneally with 500 µL of the bacterial suspension. Mice received bacterial concentration as follows: group 1: 5.01 × 10^9^ CFU/mL, group 2: 1.05 × 10^9^ CFU/mL, group 3: 5.01×10^8^ CFU/mL, group 4: 1.58 × 10^8^ CFU/mL, group 5: 5.01×10^7^ CFU/mL, and group 6: Saline, The survival, psychological and nutritional status of each mouse was monitored every 2 h after injection for 24 h, once a day after, 7 days in total. On the seventh day, all remaining mice were euthanized. During the monitoring process, the mice that died were immediately necropsied to bacterial culture and the heart, liver, spleen, lung, kidney and brain were collected. Histopathology was performed on the aforementioned organs under a light microscope (Leica DM4B optics) following 10% formalin fixation and hematoxylin–eosin staining.

## 3. Results

### 3.1. Physiological and Biochemical Characteristics of Isolate

One bacterium was isolated from heart, liver, lung, and kidney of the dead giant panda cub and revealed Gram-negative rod morphology. Additionally, this isolate growth was characterized by grey—white, circular, umbilicate colonies showing β-hemolytic after 24 h (Figure 1). Biochemical analyses showed that the bacteria were positive for oxidase, lysine decarboxylase, citrate, indole, mannitol, sucrose, maltose, V-P, and cellobiose. Collectively, and negative to others such as H_2_S production, the physiological and biochemical results indicated that this isolate belonged to *Aeromonas*.

### 3.2. Histopathological Finding of Giant Panda

Short rod-shaped bacteria were in liver, spleen, and lung. Additionally, cardiomyocyte vacuolization, degeneration and necrosis were observed, congestion and scattered necrotic occurred in liver and spleen, cells necrosis and dissolution and a lot of hemoglobin imbibition occurred in Lung, granular degeneration occurred in epithelial cells of renal tubular. There was no significant change in the umbilical cord (Figure 2).

### 3.3. Phylogenetic Analyses of the 16S rRNA and gyrB Genes of Isolated Bacterial

Amplification of 16S rRNA and *gyrB* genes revealed that the target band was 1500 bp and 1127 bp, respectively. The BLAST alignments of 16S rRNA and *gyrB* genes indicated that the isolated H, LI, LU, and K shared the same sequence and high homology with *A. veroni* (>99% homology). Therefore, this isolate was named as VGP. In addition, the phylogenetic tree based on 16S rRNA and *gyrB* sequences showed that isolate VGP were classified into the known strain species of *A. veroni* (Figure 3). Taken together, based on the physiology, biochemistry, and phylogenetic analysis, we identified isolate VGP as an *A. veronii* strain.

### 3.4. The Pathogenicity Test in Mice

The pathogenicity test of the VGP strain in mice showed post-infection (p.i), except group 6, all challenged mice exhibited clinical symptoms, such as partial loss of appetite and coarse fur. The death of mice in each group were statistically analyzed (Figure 4). The results showed that the mortality of mice in group 1 was 8 (100%), which occurred at 4 h p.i (5 died) and 6 h p.i (3 died). The mortality of 8 mice (100%) in the group 2 occurred at 6 h p.i (4 died) and 8 h p.i (4 died). The mortality of mice in the group 3 was 8 (100%), all of which occurred at 10 h p.i. A total of 7 mice died in the group 4 (87.50%), which occurred at 12, 16, 18, 20, 22 h p.i, respectively (1, 2, 2, 1, 1 mice died, respectively) and then there are no more deaths. No deaths were observed in groups 5 and 6. The lethal dosage (LD_50_) of VGP in mice was determined to be 5.14 × 10^7^ cfu by Kärber’s method. In addition, VGP was successfully re-isolated from the heart, liver, lung, kidney, and brain of the post-infected mice. 

### 3.5. Histopathology of the Inoculated Mice

Congestion is a common lesion of the liver, spleen, lung, kidney, and brain in all infected dead mice. Additionally basophilic granular aggregates consistent with bacteria were noticed in the hepatic sinuses, capsular thickening and lymphocytic necrosis were noticed in splenic. Interstitial thickening of lung were observed due to pulmonary hyperemia. Neuronal necrosis were observed in the brain and no significant lesions were observed in the heart (Figure 5).

### 3.6. Virulence Gene Analyses of the Strain VGP

In this study, we screened nine virulence genes by PCR and the results indicated that strain VGP was positive for seven of the genes (*aer*, *act*, *lip*, *exu*, *ser*, *luxs*, and *tapA*), while the remaining two genes, aha and *gcaT*, were not detected (Figure 6).

### 3.7. Antibiotic Resistance of the Strain VGP

Antimicrobial susceptibility testing results showed that strain VGP was resistant to six antibiotics, namely penicillin, ampicillin, oxacillin, amoxicillin, imipenem, and vancomycin. On the other hand, it was sensitive to 36 antibiotics, including ceftizoxime, kanamycin, and ofloxacin (Table 2), which can be used as candidate drugs for the clinical treatment of *A. veronii* infection.

### 3.8. ARGs Analyses of the Strain VGP

Among all the 96 unique ARGs, including mobile genetic elements (MGEs), a total of 31 unique ARGs were present in strain VGP. The resistance genes aac, tetA-02 and tnpA-05 were the most abundant, followed by aacC and aadA2-02. These 31 ARGs had the potential to confer resistance to a range of antibiotics, such as β-lactamase resistance (32%), aminoglycosides resistance (22%), vancomycin resistance (19%), tetracycline resistance genes (10%) and sulfonamide resistance (6%) (Figure 7).

## 4. Discussion

*A. veronii* is a pathogen that has a broad host range, infecting hundreds of aquatic animals, waterfowl, and several mammalian species [9,10,11,15]. In clinical practice, the pathogen causes human biliary sepsis and diarrhea [15]. *A. veronii* infections have been reported in a wide variety of animals; however, there have been no reports of *A. veronii* infection in giant pandas.

The physiological and biochemical characteristics of the isolate VGP showed that it was Gram-negative, β-hemolytic, and positive to oxidase, lysine decarboxylase. This is consistent with the characteristics of *Aeromonas*. Due to the wide variety of *Aeromonas* species, molecular methods are necessary to distinguish different species. 16S rRNA and *gyrB* are commonly used to identify *A. veronii* [25,29]. In this study, the phylogenetic analysis of 16S rRNA and *gyrB* genes demonstrated that isolate VGP belong to *A. veronii*.

Our study presents the first evidence that giant pandas could be infected by *A. veronii*. Histopathological analysis of captive giant panda cub showed rod-shaped bacteria in liver, spleen, and lung (This is consistent with the pathological characteristics of bacterial infection.). Then, *A. veronii* strain VGP was subsequently isolated from the heart, liver, lung, and kidney of the panda cub, that means *A. veronii* may be the causative factor of death. Furthermore, Kunming mice infection experiment results showed the time of death for Kunming mice infected with *A. veronii* was concentrated within 24 h, This indicated that strain VGP had pathogenicity. At the same time, a large number of virulence genes were detected in the strain VGP, which also proved this. The main histopathological results of infected mice were multi-organ congestion and bacterial infection. Additionally, bacteria masses were observed and *A. veronii* was isolated from the organs of Kunming mice infected with the strain VGP, this pathological feature is basically consistent with that of giant panda. Currently, the source of *A. veronii* infection in the giant panda cub is unknown and requires further investigation.

The pathogenicity of *A. veronii* is related to the expression of virulence factors [26]. One of the most important and abundant virulence factors is *Aer*, which is a cytotoxic pore-forming enterotoxin. Aer-positive *A. veronii* exhibited significantly higher mortality than Aer-negative *A. veronii* [30]. The Act gene plays a crucial role in *A. hydrophila* and significantly reduces the capacity to evoke fluid secretion [30], while *Lip* plays a common role in the pathogenicity of *Aeromonas* spp, which is secreted into the environment through the secretion system together [3]. The *pili* are well-established virulence factors for many bacteria [31]. *TapA* is mainly associated with host adherence and virulence [32], lacking *tapA* could result in a reduced ability to invade and survive within the host [33]. *Luxs* is one of the quorum sensing genes and encodes another autoinducer synthetase [34]. *Lip* plays an essential role in adhesion and integration, as well as in the pathogenesis of *Aeromonas* spp. [2]. *Ser* is an additional class of potential virulence factors related to the pathology and mortality of salmonid fish [34,35]. *Exu* is an important virulence factor that has been detected in diseased Gibel carp (*Carassius auratus gibelio*) [36]. The number of virulence genes carried by *A. veronii* may be positively correlated with its pathogenicity [2]. Our results showed that strain VGP carried seven virulence genes (*aer*, *act*, *lip*, *exu*, *ser*, *luxs*, and *tapA*). Previous research has suggested that the more virulence genes the strain carried, the smaller the LD_50_ and the stronger the pathogenicity [2], which may be related to *A. veronii’s* strong pathogenicity. As an important aquatic zoonotic agent, the potential pathogenesis of *A. veronii* should be further monitored.

Bacterial resistance to antibiotics has significant effects on animal, environmental, and human health [37]. Our results showed that *A. veronii* from the giant panda cub was resistant to six antibiotics (penicillin, ampicillin, oxacillin, amoxicillin, imipenem and vancomycin). Previous studies showed that *A. veronii* isolated from duck resistant to tetracycline, doxycycline, trimethoprim-sulfamethoxazole [8], isolated from fox resistance to tetracycline, rifampicin, penicillin, norvancomycin, bacitracin, oxacillin, clindamycin, ampicillin, novobiocin [9], Isolated from rabbit resistance to ampicillin (3/4), amoxicillin (2/4), streptomycin (4/4), tobramycin (3/4), tmikacin (2/4), gentamicin (1/4), neomycin (2/4), polymyxin B (4/4) [11], isolated from wild yak resistance to cefalexin, amoxicillin, trimethoprim-sulfamethoxazole, tetracyclin, erythromycin [12]. It can be seen that non-aquatic animal-derived *A. veronii* has universal drug resistance to ampicillin, which is contrary to the results in aquatic animals [38]. The resistance of *A. veronii* to other antibiotics is different, which may be related to the different selection and frequency of antibiotic use in different species.

ARGs are the primary reason for bacteria development of drug resistance [39]. In this study, 31 unique ARGs were positively detected in strain VGP. The β-lactam ARGs were detected which was consistent with the phenotype. At the same time, the emergence of ARGs such as *blaCTX-M-04*, *blaCTX-M-01*, *blaSHV-01*, *blaOXY* which related to extended-spectrum β-lactamase (ESBLs) should be taken seriously. Whether the emergence of these resistance genes is related to ESBL-producing *E. coli* [40] and ESBL-producing *K. pneumoniae* [41] of giant panda origin needs to be confirmed by further studies, while some ARGs showed a high abundance but there is no phenotype associated with it such as aminoglycoside resistance genes *aac*, *aaC*, *aadA2-02,* and tetracycline gene *tetA-02*, which may be related to the expression of these resistant genes.

The main lesions of the autopsy were congestion of the lung and liver. In most animals, the main symptoms of *A. veronii* infection are congestion or bleeding in internal organs, such as duck [8], fox [9], wild yaks [10], and rabbits [11]. Some will have gastrointestinal symptoms such as fox [9] and wild yaks [10]. Cold-blooded animals also show skin congestion or ulcers, etc. [3,4]. Their pathogenicity tests on mice all showed congestion or bleeding in multiple organs. This is basically consistent with our results. Congestion or bleeding of internal organs are the main clinical symptoms of animal diseases caused by *A. veronii*.

## 5. Conclusions

To our knowledge, this is the first report that an *A. veronii* could infect giant pandas. We successfully isolated the pathogen of *A. veronii* from the internal organs of a deceased captive giant panda cub by bacterial culture, with identification by PCR methods. Additionally, detection of virulence and antimicrobial resistance genes was performed by PCR methods. Our results revealed that *A. veronii* is resistant to six antibiotics, including penicillin, ampicillin and oxacillin, and carried seven virulence genes (*aer*, *act*, *lip*, *exu*, *ser*, *luxs* and *tapA*). The main pathological changes in giant panda were the large number of bacteria in multiple organs. Using artificial infection, *A. veronii* caused pathological damage to the heart, liver, spleen, lung and kidney of Kunming mice, including degeneration, necrosis, and hemorrhage. This new study reporting this potential bacterial disease in giant pandas provides insights into understanding the pathogenicity of *A. veronii* towards its host. Further epidemiological investigations and exploration of the relationship between pathogenic bacteria and the host are needed.

## Figures and Tables

**Figure 1 animals-13-02779-f001:**
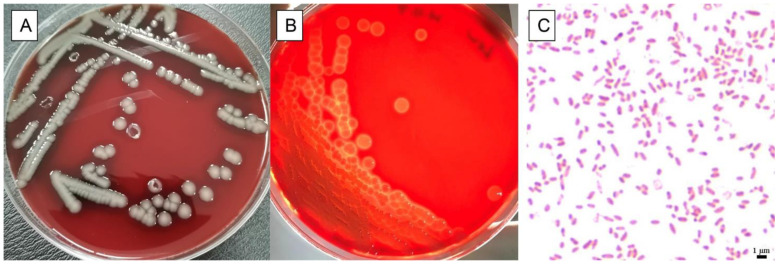
Physiological characteristics of isolated isolate. Growth of isolate on sheep blood plates showed grey-white, circular, umbilicate colonies (**A**), and β-hemolysis (**B**). Gram staining of isolate showed typical Gram-negative, slightly curved, single or in pairs (**C**).

**Figure 2 animals-13-02779-f002:**
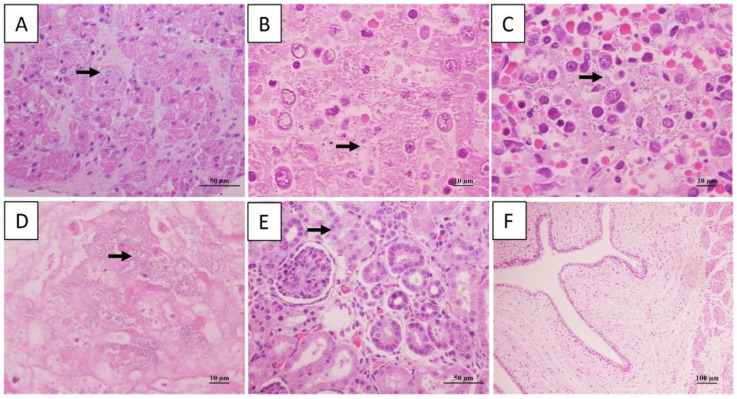
Histopathological changes in dead giant panda. Cardiomyocyte vacuolization in heart (**A**). Rod-shaped bacteria and necrosis in hepatic (**B**), and spleen (**C**). Exudates with bacterial masses in the alveolar space (**D**). Granular degeneration in epithelial cells of renal tubular (**E**). No significant changed in umbilical cord (**F**). The arrow represents the location of the lesion.

**Figure 3 animals-13-02779-f003:**
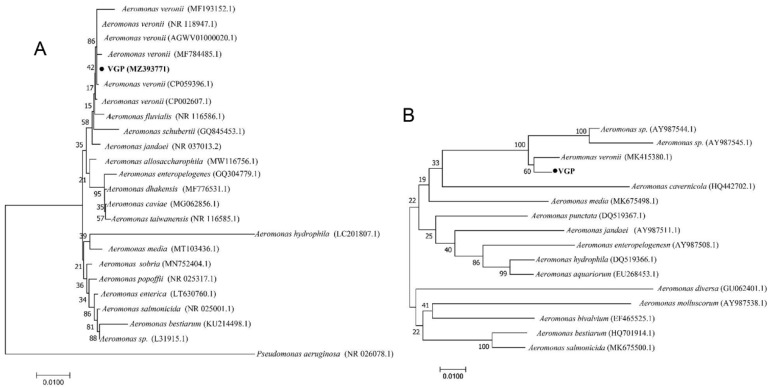
Neighbor-Joining phylogenetic tree generated based on 16S rRNA (**A**) and *gyrB* (**B**) gene of the strain VGP and other sequences of *A. veronii* isolates detected in the present study and the other *Aeromonas* spp. from Genbank. *Pseudomonas aeruginosa* was used as an outgroup species. Bootstrap values out of 1000 repetitions were indicated above each branch. The scale bar represents 0.01-nucleotide change per nucleotide position.

**Figure 4 animals-13-02779-f004:**
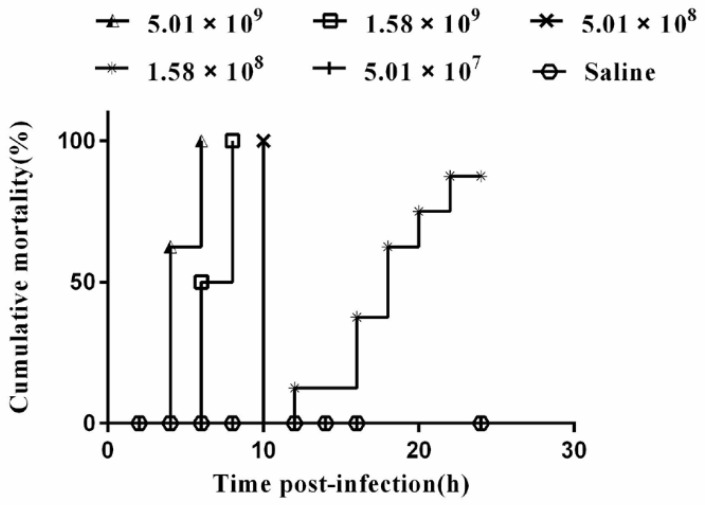
The cumulative mortality of Kunming mice (n = 48) infection strain VGP in 24 h. Each line represents a different concentration of bacterial. The experiment was observed for 7 days. Since deaths do not occur after 24 h p.i, only deaths within 24 h p.i are shown.

**Figure 5 animals-13-02779-f005:**
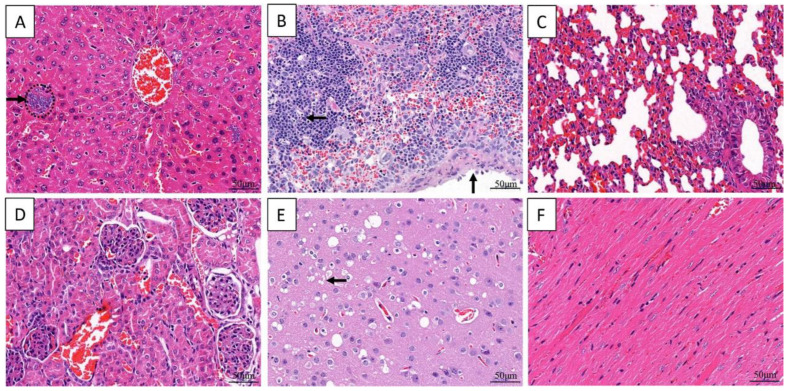
Histological changes in the organs of Kunming mice inoculated with the strain VGP. Basophilic aggregates of bacteria (arrow) and congestion in the sinuses (**A**). Lymphocyte necrosis (horizontal arrow) and capsular thickening (vertical arrow) in spleen (**B**). Congestion in lung (**C**) and in kidney (**D**). Neuronal necrosis and congestion in brain (**E**). No significant changes in heart (**F**).

**Figure 6 animals-13-02779-f006:**
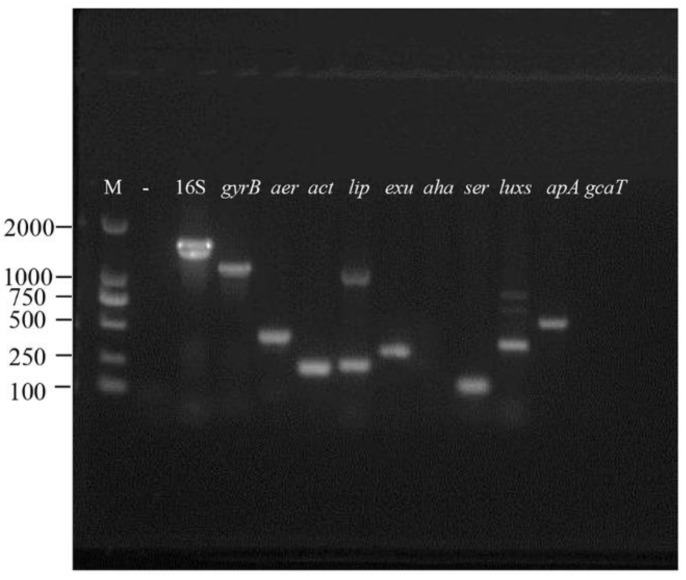
Agarose gel electrophoresis of nine virulent genes (*aer*, *act*, *lip*, *exu*, *aha*, *ser*, *luxs*, *tapA*, *gcaT*) and house-keeping genes (16S rRNA and *gyrB*) in the strain VGP. M: DNA molecular weight standard; “-”: negative control.

**Figure 7 animals-13-02779-f007:**
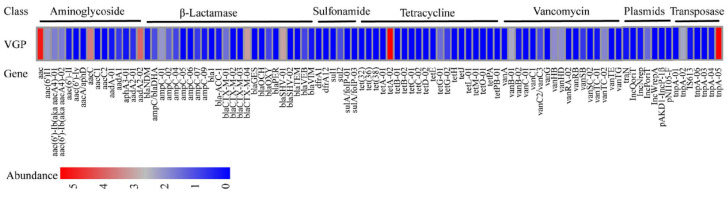
Antimicrobial resistance genes abundance in strain VGP.

**Table 1 animals-13-02779-t001:** Primer pairs utilized for virulence gene detection.

Target Gene	Primer Sequence (5′-3′)	Product Size (bp)	Tm (°C)
*aer*	F: CCTATGGCCTGAGCGAGAAG	431	56
	R: CCAGTTCCAGTCCCACCACT		
*act*	F: GAGAAGGTGACCACCAAGAACA	232	60
	R: AACTGACATCGGCCTTGAACTC		
*ser*	F: CTCCTACTCCAGCGTCGGC	128	64
	R: GATCGTCGGTGCGGTTGT		
*aha*	F: GGCTATTGCTATCCCGGCTCTGTT	1082	60
	R: CGGTCCACTCGTCGTCCATCTTG		
*lip*	F: CACCTGGTKCCGCTCAAG	247	56
	R: GTACCGAACCAGTCGGAGAA		
*exu*	F: AGACATGCACAACCTCTTCC	323	56
	R: GATTGGTATTGCCYTGCAA		
*luxs*	F: GATCCTCTCCGAGGCGTGG	369	58
	R: AGGCTTTTCAGCTTCTCTTCC		
*tapA*	F: ATGACCTCTAGCCCCAATA	550	52
	R: ACCCGATTGATTTCTGCC		
*gcaT*	F: CTCCTGGAATCCCAAGTATCAG	237	55
	R: GGCAGGTTGAACAGCAGTATCT		

**Table 2 animals-13-02779-t002:** Antibiotic resistance of the strain VGP.

Antibiotics	Content (/disc)	Inhibition Zone (mm)	Sensitivity
PEN	10 µg	0.00	R
PIP	100 µg	24.96	S
AMP	10 µg	0.00	R
OX	1 µg	0.00	R
AMX	20 µg	0.00	R
MOX	30 µg	36.05	S
CAZ	30 µg	30.02	S
CFM	5 µg	40.22	S
CMZ	30 µg	35.76	S
FEP	30 µg	34.96	S
CTX	30 µg	39.06	S
CA	30 µg	29.20	S
CZ	30 µg	19.66	S
CTR	30 µg	42.32	S
FOX	30 µg	31.98	S
TZD	100/10 µg	28.44	S
CXM	30 µg	32.22	S
CEC	30 µg	29.60	S
AMS	10/10 µg	12.26	I
CFP	75 µg	33.92	S
ZOX	30 µg	35.76	S
AT	30 µg	36.68	S
MEM	10 µg	29.84	S
IPM	10 µg	19.40	R
K	30 µg	22.44	S
GM	10 µg	22.46	S
S	10 µg	18.49	S
ENX	10 µg	17.54	I
OFX	5 µg	29.54	S
NOR	10 µg	22.45	S
LOM	10 µg	25.59	S
FOX	5 µg	21.02	S
LVX	5 µg	27.48	S
CIP	5 µg	26.86	I
GAT	5 µg	30.24	S
C	30 µg	32.88	S
VA	30 µg	0.00	R
AZM	15 µg	28.11	S
DX	30 µg	12.83	I
MI	30 µg	18.95	S
SXT	25 µg	20.91	S
TMP	5 µg	19.88	S

Note: PEN, penicillin; PIP, piperacillin; AMP, ampicillin; OX, oxacillin; AMX, amoxicillin; MOX, moxalactam; CAZ, ceftazidime; CFM, cefepime; CTX, cefotaxime; CA, cephalexin; CZ, cefazolin; CTR, ceftriaxone; FOX, cefoxitin; TZD, piperacillin/tazobactam; CXM, cefuroxime; CEC, cefaclor; AMS, ampicillin/sulbactam; CFP, cefoperazone; ZOX, ceftizoxime; AT, aztreonam; MEM, meropenem; IPM, imipenem; K, kanamycin; GM, gentamicin; S, streptomycin; ENX, enoxacin; OFX, ofloxacin; NOR, norfloxacin; LOM, lomefloxacin; FOX, fleroxacin; LVX, levofloxacin; CIP, ciprofloxacin; GAT, gatifloxacin; C, chloramphenicol; VA, vancomycin; AZM, azithromycin; DX, doxycycline; MI, minocycline; SXT, trimethoprim-sulfamethoxazole; TMP, trimethoprim. S, sensitive; I, intermediary; R, resistant.

## Data Availability

All data generated or analyzed during this study are included in this published article and its Appendix A files.

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
