# Peer review of "First Isolation and Identification of Aeromonas veronii in a Captive Giant Panda (Ailuropoda melanoleuca)"

_animals, 2023, doi:10.3390/ani13172779_

Round 1

Reviewer 1 Report

The authors found a case in giant panda infected by Aeromonas veronii and analyzed the morphological and biochemical features as well as 16S rDNA and gyrB gene sequences of the pathogen. The result will help giant panda keepers to find methods to prevent and cure the case. The authors should revise the manuscript before publication.

Major comments:

1)      Aeromonas veronii is facultative anaerobic bacterium and widely distributed in the aquatic environment. How did the authors judge that giant panda can be infected by A. veronii and the bacteria was not contained by the environment around the giant panda? What is the abundance of the bacteria in internal organs of the panda? What is symptom for the dead panda like the symptom other animals infected by the bacteria?

2)      Aeromonas veronii is widely distributed in aquatic environment and caused infections in many animals. The authors should compare the difference of the pathogenic bacteria hosted in giant panda with other animals, which is important to identify the specificity.  

Little comments:

1)      Simple Summary: “A. veronii (strain VGP), which is considered a dominant infecting strain”. How to identify a dominant infecting strain in giant panda?

2)      How to get chromosomal DNA (2.6. Molecular Identification of Antibiotic Resistant Genes (ARGs)) for the bacteria?

Author Response

Dear Reviewers,

Thank you for your valuable suggestions to make our article more perfect, we have corrected the shortcomings you pointed out. Peer-to-peer replies are added to the attachment.

Reviewer 2 Report

Abstract

Line 14: “…the infection ofàwith strain VGP…”

Line 17: “…biological characteristics” (not capital letter)

Line18-21: Explaining the aim of the method right after the methods name would make it easier for the reader, e.g. : “Here, we use 16S rDNA and gyrB genes for bacterial identification, disk diffusion method for phenotypic antimicrobial susceptibility testing, smart chip real-time PCR for genotyping, multiplex PCR for determination of virulence genes and acute toxicity test in mice for testing the pathogenicity of the isolate.” (be aware that biochemical tests for species identification have been performed in the study, which is missing in this abstract)

Line 21-22: “The isolate was identified as A. veronii strain VGP based on the biochemical properties and genetic analysis”

Line 23-24: Recommended: “We found that the strain carried 31 antibiotic resistance genes and revealed antimicrobial resistance phenotypically to several antibiotics including …..and carried virulence genes including….” It´s recommended to use the generic names of the antimicrobial substance not the product names, e.g.: amoxicillin and vancomycin (not amocillin and vancocin)

Line 27 and 29: “infection with strain VGP…”

Line 29: “..VGP was the leading cause of death…” (there aren´t several leading causes. There is either one leading cause of death or there is one of the causes of death.

Introduction

Line 34: “…in the aquatic environment and causes gastroenteritis, …”

Line 36: “common pathogen in aquatic animals including …” (avoid unnecessary repetition)

Line 39: “…clinical symptoms are skin ulcers and visceral hemorrhages in these species.”

Line 41: This sentence contains the further effected species, which doesn´t need a new paragraph. And regarding the information in this sentence: is A. veronii pathogenic for snake eagle and scarlet ibis? Indicate accordingly (infection is not equal to disease).

Line 43: The sentence “Wild waterfowl may carry pathogenic Aeromonas species in their intestine…” disturbs the composition and the flow of the previous and following sentences, where the effected species and the clinical signs are described. But this sentence gives the information about the spread of the bacterium. Therefore, it would be more suitable if this comes after the sentence: “In addition, infection ….” In the line 48.

Line 55: Klebsiella pneumoniae (“i” is missing).

Line 57: What is meant with “gradually highlighted”? Is it occurring more frequently in the giant panda? Is it more frequently reported?

Materials and methods

Line 63: “Organ sampling and bacterial isolation” is recommended title.

Line 64-65: Recommended style: “A captive one-day-old giant panda cub died in Sichuan Province, China in 2020 without any sign of physical trauma in the postmortem examination. Heart, liver…were collected for bacterial examination” (were there signs of septicemia, hyperemia, hemorrhages? Why was bacterial analysis conducted? If so, it would be good to mention).

Line 66-67: Recommended format of the sentence: “The samples were cultivated on 5% (?) sheep blood agar at 37°C for 24h and the dominant uniform bacterial colonies were purified by…..”

Line 70: “Strain” is a term that you can use only after genomic analysis, therefore, “isolate” would be recommended here.

Line 71: “named as (?) VGP”. What is VGP? It would be useful to indicate the abbreviations on the first usage.

Line 73: What is “AGP20-12”? It has never been mentioned before and would be hard to understand for people who are not familiar with the term. Please indicate the abbreviations on the first usage.
“studied by(?) bacterial fluid”. Do you mean “bacterial suspension was gram-stained and examined under the light microscope?

Line 74: As it is a core practice to use sterile material in bacterial cultures, the sentence here is not necessary and makes the reader think “were the previous steps done by a non-sterile loop?”. 
Side information to this sentence, that should be deleted: “loop” is common word for “inoculation ring”.
Remark to biochemical test: Which biochemical tests did you apply, trehalose, sorbitol, lactose etc.? API Strips? Please indicate accordingly.

Line 81-84: These primers are not shown in the table 1? It’s just virulence genes present in the mentioned table. So, either show these primers in detail in the table including product size and base sequence and just refer to the table or you mention it in detail here as it is now and don’t refer to table.

Line 84-85: Please delete “The amplified products….” sentence as it is the routine thing to do during gene analyses and not needed to mention extra. Recommended version: “The sequences analysis was done by blasting in NCBI…”
Please indicate that these two genes were amplified by classic PCR.

Line 89-93: Recommended formulation of this part sentence: “Virulence genes including: … were detected by classic PCR using primers in previously published studies. The PCR Products were sequenced by Sangon Biotech…. and sequences were analyzed by blasting in…”.

Line 95: Recommended title for the table: “Primer pairs utilized for virulence gene detection” (if 16S and gyrB primers will be presented here the title should be adjusted accordingly.

Line 96: Recommended title term to use for “Antibiotic resistance test” is: “Antimicrobial susceptibility testing”. Please adjust this in the remaining parts of the text.

Line 98-98: Based on which criteria are these antibiotics tested?

Line 100: amoxicillin (“x” is missing here)

Line 110: “according to the CLSI 2020 breakpoints”

Line 111: “…antibiotic resistance genes” (not resistant genes)

Line 124: Recommended: “Pathogenicity testing in Kunming Mice”. As the bacterial isolate is not a toxin or a substance, it should be more suitable to talk about “pathogenicity”. But if this is a common term for just purpose, my comment can be ignored.

Line 128-132: “Bacterial isolates were incubated…and following the suspension in 0.9% endotoxin-free saline solution, animals were inoculated intraperitoneally with 500 µL of the bacterial suspension. Mice received bacterial concentration as follows: Group 1….”

Line 132-134: Clinical signs and survival were observed every 2hours within the first 24 hours and 7days in total” (how frequent was the observation after the first 24hours? Please indicate accordingly.

Line 135: “lesions in the organs” (not of the organs)

Line 136-139: Recommended: “…, all remaining mice were euthanized, and bacterial culture and histopathology were performed from the organs including….”

Line 140-147: As it is part of the pathogenicity testing, a separate: “Histopathology” title is not suitable. This should be added to the previous section and recommended is: “ Histopathology was performed on the aforementioned organs under a light microscope (Leica DM4B optics) following 10% formalin fixation and hematoxylin-eosin staining.

Results:

Line 149: “Physiological and Biochemical characteristics of the Isolate”

Line 150-151: Here again: it is not correct to talk about strains. And it is also not realistic that you isolate several strains from the same animal, that you claim as the cause of death. “Gram” is a staining method and the way the sentence is arranged makes one almost think gram is an analysis like PCR…Recommended: “bacterium was isolated from heart,… and revealed gram-negative rod morphology.”

Line 152-154: Please make sure to use “isolate”, not “strain”. Recommended: “growth was characterized by grey-white, circular, umbilicate colonies showing β-hemolysis after 24 h”.

Line 159-162: from the picture A one can see only the hemolysis. The remaining description about the colonies therefore is not corresponding to the picture. Please adjust the figure legend accordingly. To the legend of Figure B: “gram-negative, slightly curved rods in pairs or solitary”.

Line 175: “Pseudomonas” (“e” is missing)

Line 178: “Pathogenicity Test in Mice”

Line 179-182: The purpose of the work is already mentioned and should be repeated here. This is the results section. Please indicate clearly how many mice died within the first 24hours. Otherwise, the rest of this part wouldn’t make sense. If all the animals died within 24 hours, of course there won’t be any change to mortality rates after that.

Line 183-185: The mortality information here doesn’t make sense: How can group 1 show 100% mortality within 6h p.i.? This is the zero group that shouldn’t have received any bacterial inoculation!? Either the explanation about the groups is given wrongly in the pervious M&M section or there is an error here. And how can the LD50 be 5.14 × 107, given that the group 6 receiving 5.01×107 cfu/mL showed no mortality?

Line 188-189: Mortality in mice cannot prove the mortality in a completely different species. Especially considering the totally different routes of infection as the panda didn’t receive it intraperitoneally.

Line 191: n=8 per experiment group. So, if total n is referred: n=48

Line 194: “Histopathology of the inoculated mice”

Line 195-196: Please skip the first sentence in this section, as it is not a result.

Line 196-198: Recommended: “Infected mice revealed common lesions in the liver, spleen, lung, kidney, and brain. Capsular thickening and parenchymal congestion in the liver and spleen were observed.”

Line 199: “Additionally basophilic granular aggregates consistent with bacteria were noticed in the hepatic sinuses and lymphocytic necrosis”.

Line 201: Please describe the inflammation in the lung and kidney more precisely: interstitial? Parabronchial? Peritubular? Inflammatory cell type? What one can see from the picture below is only interstitial thickening due to mostly pulmonary hyperemia.

Line 202-204: Recommended: “neuronal necrosis and congestion were observed in the brain and no significant lesions were observed in the heart.”

Figure 4 Legend: Recommended: “Histological changes in the organs of Kunming mice inoculated with the isolate VGP. Basophilic aggregates of bacteria (arrow) and congestion in the sinuses (A). Lymphocyte necrosis (horizontal arrow) and capsular thickening (vertical arrow) in spleen (B). Congestion in lung (C) and in kidney (D). Neuronal necrosis and congestion in brain (E). No significant changes in heart (F).” (Be aware that in pathology language congestion refers to “increased amount of blood in the vessels”, so, “blood congestion” is a redundant expression.

Line 210: Strain is not equal to isolate.

Figure 5 Legend: Recommended: “Agarose gel electrophoresis of nine virulence genes (aer, act, lip, exu, aha, ser, luxs, tapA, gcaT) and house-keeping genes (16S rDNA and gyrB) in the isolate VGP. M: DNA molecular weight standard, “-”: negative control

Line 218-219: Please skip the first sentence, which doesn’t give any specific information.

Line 221: What is the reason that you mention ceftizoxime, kanamycin and ofloxacin explicitly? Are they the drugs commonly in use in zoo environments, are they the substances that are commonly used for Aeromonas sp.? Indicate accordingly.

Line 224+230: “amoxicillin” (not amocillin), “vancomycin” (not vancocin)

Line 231: “I, intermediary” (not “M, moderately susceptible”: there isn’t any M in the table!)

Line 234: “Mobile Genetic elements” (not “movable”)

Line 235: “…ARGs were present in strain VGP…”

Line 235-6: “…tetA-02…” (not tetA-01)

Figure 6 Legend: “Antimicrobial resistance genes abundance in strain VGP”. (It is clear what the colors indicate based on the scale at the lower left of the figure).

Discussion:

Line 250-252: No where in the M&M or results was mentioned that histopathology was done in the organs of this panda and here suddenly there is description of histological lesions! According to the previous text, the only histopathological analysis was done in the organs of the inoculated mice. Please clarify this point.

Line 254-257: Be cautious about interpretation of two separate things: the morbidity in mice following intraperitoneal inoculation doesn’t prove the morbidity in panda cub, which there hasn’t been any evidence of the infection route. If histopathology was really done in the organs of the panda and if bacteria were really observed in the alveoli in the lungs, this indicates an aspiration either perimortem or antemortem, which must be supported by corresponding inflammatory reaction!

Line 259-264: Please edit this part in terms of language. There are many spots where the sentence is incomplete or wrongly arranged.

Line 286-291: Language and sentence construction need to be edited here too. Based on these results there is nothing outstanding/unusual for this isolate of A. veronii compared to any other A. veronii, which doesn’t add up/change the treatment options in pandas. Please discuss this part based on the intrinsic/expected resistances of A. veronii.

Line 293-298: please revise the language and discuss the genotypic and phenotypic resistance of the isolate based on the known information on A. veronii. Are there differences, unusual observations etc. “significant” is a term that one can use only after statistical analysis, which is not the case here.

Line 300-308: The results of the mice experiment doesn’t tell much about the situation in the panda.

Line 312: It is not possible to isolate a bacterium by PCR! Please revise the sentence. What has been done in this study is detection of virulence and antimicrobial resistance genes by PCR and identification of the bacterial isolate by PCR followed by sequence analysis. Please express these different things precisely and correctly.

The text in general is clear to understand and well written. However there are repetative errors in the general usage of some words and construction of sentences.

Author Response

(The authors gave the same response as above.)

Reviewer 3 Report

In the submitted manuscript (Manuscript ID: animals-2469620), the authors identified A. veronii from multiple organs of a dead captive giant panda cub. And, they also found that the strain carries resistance genes and virulence genes. Toxicity test of Kunming mice showed that the infection of strain VGP may be the cause of the death of the giant panda cub. Overall, it is a remarkably comprehensive study, providing useful information for panda disease. According to Koch’s Postulatesthey presents the first evidence that giant pandas can be infected by A. veronii. For the benefit of the reader, some points need modifying.

Specific:

1.     16S rDNA should be 16S rRNA 

2.     modify cfu/mL to CFU/mL

3.   make clear group 1-5 in section 2.7 and 3.3

4.  A. veronii could be italic

 Minor editing of English language required.

Author Response

(The authors gave the same response as above.)

Round 2

Reviewer 1 Report

no more comments

Author Response

Dear reviewer,

We would like to thank the reviewers for constructive suggestions revising the manuscript. The suggestions have helped us to improve our study.

Kind regards,
Xiaoyan Su

Reviewer 2 Report

Presentation of this work in a different context is recommended. Based on the severely autolytic state and lacking consistent inflammation in the tissues of this panda cub, the evidence of causality is very week. But this is still a good evaluation of an isolate from a panda and this work can still be presented with a different focus and in a journal having bacteriology interest.

There are still several and repetative errors in the language.

Author Response

Dear Reviewer,

We would like to thank the reviewer for constructive suggestions revising the manuscript. The suggestions have helped us to improve our study. The responses to your comments was in an attachment.

Thank you!
